# Real-World Comparison of Health-Related Quality of Life Associated with Use of Immune-Checkpoint Inhibitors in Oncology Patients

**DOI:** 10.3390/jcm13164918

**Published:** 2024-08-20

**Authors:** Abdulrahman Alwhaibi, Miteb A. Alenazi, Saad D. Alnofaie, Abdullah M. Aldekhail, Rakan J. Alanazi, Sultan Alghadeer, Abdulrhman A. Alghamdi, Saleh A. Alanazi

**Affiliations:** 1Department of Clinical Pharmacy, King Saud University, Riyadh 11451, Saudi Arabia; 444106458@student.ksu.edu.sa (A.M.A.); salghadeer@ksu.edu.sa (S.A.); 2Department of Pediatrics, College of Medicine, King Saud University, Riyadh 11451, Saudi Arabia; mitalanazi@ksu.edu.sa; 3Information Technology, King Abdulaziz Medical City, Ministry of National Guard Health Affairs, Riyadh 11426, Saudi Arabia; nofaies@mngha.med.sa; 4Pharmacy Practice Department, College of Pharmacy, Alfaisal University, Riyadh 11533, Saudi Arabia; rjalanazi@alfaisal.edu; 5Pharmaceutical Services Department, Prince Sultan Military Medical City, Riyadh 11159, Saudi Arabia; aaalghamdi@psmmc.med.sa; 6College of Pharmacy, King Saud bin Abdulaziz University for Health Sciences, Riyadh 14611, Saudi Arabia; anazis@ngha.med.sa

**Keywords:** health, quality of life, immune checkpoint inhibitors, atezolizumab, pembrolizumab, nivolumab, durvalumab

## Abstract

**Background:** Immune checkpoint inhibitors (ICIs) offer a new treatment approach for cancer, with an improvement in patient survival. However, it remains unclear whether their use impacts the quality of life of treated patients. This study aims to compare the health-related quality of life (HRQoL) of patients treated with different anti-PD-1 and anti-PD-L1 drugs, including several single or combination therapies. **Methods:** This is a prospective observational study conducted with adult cancer patients who received at least one dose of anti-PD-1 or anti-PD-L1. The HRQoL of all adult patients was assessed using the European Organization for Research and Treatment of Cancer (EORTC) Quality of Life Questionnaire-Core 30 module (QLQ-C30), version 3, Arabic version. **Results:** A total of 199 patients were found to be eligible for this study. Of these, 93 patients (82 on a single medication and 11 on multiple ICIs) completed the questionnaire, with a response rate of 46.7%. The majority of patients were treated with pembrolizumab (39.8%), followed by a smaller number treated with nivolumab (35.5%). Most of the patients were diagnosed with solid and advanced malignancies—88.2% (*p* = 0.023) and 87.1% (*p* = 0.021), respectively—with a significant difference between treatment groups. The median functioning score was 84.7%, with no significant difference between treatment groups (*p* = 0.752). Fatigue and pain were noted in >50% of patients, influencing the overall cohort’s score related to these symptoms, with scores of 88.8% and 83.3%, respectively. Although a non-significant variation was found in the scores of all combined symptoms among all groups, ranging from 82.1% to 90.4% (*p* = 0.931), patients receiving anti-PD-1 + anti-PD-L1 tended to more frequently complain about fatigue, pain, dyspnea, and constipation and hence, exhibited the worst, yet non-significant, scores compared to those of the other groups, with *p* = 0.234, *p* = 0.79, *p* = 0.704, and *p* = 0.86, respectively. All combined groups scored 83.3% on the global health scale. Nevertheless, the nivolumab-treated patients scored 75%, which was the worst global health score compared with those of the other groups, but this score was not statistically significant (*p* = 0.809). **Conclusions:** Our findings revealed no significant difference in the impact of different ICIs on the HRQoL of cancer patients. However, a larger number of cases would be necessary to provide a robust analysis and to yield conclusive results.

## 1. Introduction

Immune checkpoint inhibitors (ICIs) are novel antineoplastic therapies that revolutionized treatment of cancer in 2011 after the approval of anti-cytotoxic T lymphocyte antigen-4 (anti-CTLA-4), ipilimumab, for the treatment of melanoma [1]. Several ICIs have been developed since then, including anti-programmed cell death 1 ligand (anti-PD-L1) [atezolizumab, avelumab, durvalumab] and anti-programmed cell death receptor 1 (anti-PD-1) [nivolumab, pembrolizumab, cemiplimab] [2,3]. CTLA-4 and PD-1 are inhibitory receptors presented by T-cells, while PD-L1 is a ligand protein highly expressed by tumor cells, binding to PD-1. The activation of signaling pathways downstream of CTLA-4 and PD-1 mediates the suppression of T-cell immunosurveillance activities. Thus, the use of ICIs interrupts the inhibitory signaling and potentiates T-cells, which results in cancer cell identification and elimination. Given their clinical ability to prolong the survival of cancer patients, including those with a poor prognosis [4], ICIs have become the standard of care for the treatment of various types of cancer, such as non-small cell lung cancer (NSCLC), metastatic melanoma, metastatic Merkel cell carcinoma, small-cell lung cancer, breast cancer, hepatocellular carcinoma, head and neck squamous cell cancer, Hodgkin’s lymphoma, urothelial carcinoma, bladder cancer, renal cell carcinoma, and subtypes of metastatic colorectal cancer [2,4]. Overall, cancer treatment using ICIs appears promising, and they are expected to become clinically appropriate for other types of cancer in the near future [4].

Quality of life (QOL) is defined as “the degree to which an individual is healthy, comfortable, and able to participate in or enjoy life events” [5]. It is a subjective perception arising from self-evaluation of multiple domains, including the physical, functional, emotional, social, professional, and financial well-being of the person [6,7,8]. Despite having overall survival and progression-free survival as the focus of studies investigating cancer therapies, an evaluation of health-related quality of life (HRQoL) during and post-treatment has been adopted in recent studies, as it could influence disease progression and patient survival [9,10,11]. Consequently, patient-reported outcome measures (PROMs) are currently recognized as essential outcomes in clinical trials, particularly for those conducted in the oncology field [12,13]. With the widespread use of ICIs, assessing the HRQoL associated with their use is of great importance in order to understand the prognostic implications of their use for cancer patients [14].

Several studies have investigated the HRQoL associated with the use of ICIs, comparing HRQoL among cancer patients treated with either ICIs or other treatment modalities. A study of patients with recurrent head and neck carcinoma who were treated with either nivolumab or a single standard therapy (methotrexate, docetaxel, or cetuximab) showed significantly deteriorated physical, role, social functionality, pain, and sensory symptoms with standard therapy, as reflected by the European Organization for Research and Treatment of Cancer (EORTC) Quality of Life Questionnaire-Core 30 module (QLQ-C30) and Quality of Life Questionnaire—Head and Neck module (EORTC QLQ-H&N35) assessment at both week 9 and 15, compared to the results for the nivolumab group [15]. Another study assessing patients with advanced gastric/gastroesophageal junction cancer or esophageal adenocarcinoma, treated with either nivolumab plus chemotherapy or chemotherapy alone, observed stable or improved HRQoL in the former group, with a decrease in undesirable symptoms compared to the results for chemotherapy alone, according to the EuroQol instrument-5 dimensions (EQ-5D) and the Functional Assessment of Cancer Therapy-Gastric (FACT-Ga) analysis [16]. A 7-year follow-up of a randomized controlled trial assessing the effects of fotemustine, ipilimumab plus fotemustine, or ipilimumab plus nivolumab in melanoma patients with asymptomatic brain metastases showed preserved HRQoL from baseline to week 12 in all treatment arms [17]. Despite this fact, the ipilimumab plus nivolumab group exhibited the lowest change in functioning scores, based on the EORTC QLQ-C30, and the highest improvement in Quality of Life Questionnaire—Brain Neoplasm Module (EORTC QLQ-BN20) score. Unlike previous studies in which nivolumab showed maintained or improved in HRQoL scores, Motzer, R.J. et al. reported no significant difference in the quality of life in advanced renal carcinoma patients treated with nivolumab vs. everolimus, based on the Functional Assessment of Cancer Therapy–Kidney Symptom Index Disease-Related Symptoms questionnaire (FKSI-DRS), at baseline or at 104 weeks post-treatment, despite a numerical improvement in the FKSI-DRS scores among the nivolumab group in the latter assessment [17]. Vaughn, D.J. et al. evaluated the HRQoL of patients with advanced urothelial cancer over a 51-week period post-treatment with pembrolizumab vs. chemotherapy (paclitaxel, docetaxel, or vinflunine) [18]. Interestingly, although no difference at baseline was noted between the two groups based on the EORTC QLQ-C30 score, a significant worsening in global health status, reflected by a change in the mean score from the baseline, was observed in the chemotherapy arm at week 15 compared to that for pembrolizumab. More importantly, a continuous worsening in HRQoL was observed over time (week 15 to week 51) in the chemotherapy-treated patients, while a stable health status, or even an improvement in symptoms, particularly constipation, was found in the pembrolizumab group. Another study comparing the HRQoL results for pembrolizumab monotherapy vs. standard of care (platinum-based chemotherapy) in patients diagnosed with advanced non-small-cell lung cancer (NSCLC) using the EORTC QLQ-C30 and Quality of Life Questionnaire—Lung Cancer Module (EORTC QLQ-LC13) revealed a maintained or significant improvement at week 15 from treatment initiation with pembrolizumab to a greater degree than that from platinum-based chemotherapy [19]. The same finding was reported in another trial comparing the pembrolizumab–pemetrexed–platinum regimen to pemetrexed–platinum treatment alone in patients with untreated, metastatic, non-squamous NSCLC, in which patients treated with the former regimen maintained better global health and quality of life at week 21, based on EORTC QLQ-C30, EORTC QLQ-LC13, and EQ-5D scores, when compared with the results for the other regimen [20]. Finally, a randomized controlled trial investigating the outcome of atezolizumab plus chemotherapy with/out bevacizumab vs. bevacizumab plus chemotherapy in NSCLC patients revealed no significant difference in global health, physical functioning, and symptoms burden during the induction and maintenance phase of the treatment arms, according to EORTC QLQ-C30 and QLQ-LC13 scores [21].

It is worth noting that all aforementioned findings were reported exclusively in clinical trial settings, with limited comparison between ICIs. Another level of evidence comparing between ICIs or between different time points of the same ICI exists in the literature. A cross-sectional, chart-based review study evaluating the HRQoL of melanoma patients, off-treatment (single anti-PD-1 vs. nivolumab/ipilimumab) for at least 6 months with no recurrence, revealed worse overall health with combination therapy vs. single anti-PD-1 therapy, based on the EORTC QLQ-C30 and Quality of Life Questionnaire—Fatigue Module (EORTC QLQ-FA12) scores [22]. In spite of this, a significantly higher number of patients treated with single anti-PD-1 reported tension compared to that for the group treated with the combination therapy. Another retrospective study conducted using individuals who reported about HRQoL post-ICI on social media using the global health domain of EORTC QLQ-C30 and the Functional Assessment of Cancer Therapy—General (FACT-G) questionnaires showed that the majority (54.8% of 115 participants) of patients had poor global health, while 45.2% perceived it as stable or good [23]. In a single-arm prospective study investigating the HRQoL in melanoma patients treated with ipilimumab at baseline and 10–12 weeks later revealed significantly better global health, role and social functioning, and fatigue levels among patients who completed the EORTC QLQ-C30 at baseline and 10–12 weeks later compared to the results for those completing the questionnaire at baseline only [13]. Nevertheless, an overall deterioration in global health, physical functioning, role functioning, fatigue, and appetite was generally observed among all participants, which was significantly and clinically meaningful. In a cross-sectional study using the EORTC QLQ-C30, in addition to other questionnaires, among melanoma patients receiving nivolumab as an adjuvant therapy, a numerical and temporary improvement in physical, role, cognitive, and social functioning, as well as global health, was observed in the first 6 months of the course of treatment compared to the results for patients who had finished treatment or those who had been on treatment > 6 months [24]. In another prospective study evaluating the HRQoL of metastatic urothelial carcinoma patients treated with pembrolizumab, no significant change was found in the total score of FACT-G at baseline, after 3 months, and after 6 months of therapy; hence, no change in the HRQoL was noted throughout the duration of therapy [25].

In addition to the scarcity of research globally and the conflicting evidence regarding HRQoL associated with the use of ICIs in cancer treatment, no study evaluating the HRQoL in cancer patients treated with ICIs in Saudi Arabia exists in the literature. Our study aims to prospectively assess the impact of different ICIs on the HRQoL of cancer patients using the EORTC QLQ-C30, version 3, Arabic version.

## 2. Methods

This was a prospective, questionnaire-based study of adult patients diagnosed with cancer, admitted to the oncology center at King Khalid University Hospital (KKUH) and National Guard Hospital (NGH) between 2008 and 2022, who received at least one dose of any ICI in both hospitals. The institutional review board approval was received from both hospitals (IRB Project No. E-21-6204 on 14 August 2023; NRC23R-152-02 on 16 May 2023). Electronic medical records were retrieved and reviewed for demographic data, clinical diagnoses, and comorbidities prior to ICI initiation. ICI protocols used for treatment were documented as well. Patient quality of life was assessed using the EORTC QLQ-C30, version 3 [26], Arabic version. Permission to use the questionnaire was obtained and approved through a website [27]. Patients were contacted, and their consent was obtained for participation in the study.

The questionnaire consists of multi-item and single-item scales [26]. Multi-item scales include: functional scales, assessing five different domains—physical (5 questions), role (2 questions), cognitive (2 questions), emotional (4 questions), and social (2 questions); symptom scales, assessing three symptoms—pain (2 questions), fatigue (3 questions), and nausea and vomiting (2 questions); and one scale to assess global health (2 questions). The rest are single-item scales assessing symptoms commonly reported by cancer patients (dyspnea, appetite loss, sleep disturbance, constipation, and diarrhea) and financial impact attributed to the physical condition or disease treatment (1 question each). With the exception of items assessing global health (questions 29 and 30, respectively), each item is scored 1 to 4: 1—Not at all, 2—A little, 3—Quite a bit, 4—Very much. Because of the way in which the questions are worded, a higher score resulting from each question indicates a worse level of functioning and symptoms. Regarding global health questions, responses range from 1 (Very poor) to 7 (Excellent), such that a higher score indicates better health.

For an easier and linear interpretation of the results, the scores were transformed into scales ranging from 0–100, using the EORTC QLQ-C30 scoring manual [28]. In detail, for functional assessment, when the sum of the raw scores for physical, role, cognitive, emotional, and social functioning were calculated separately, they were incorporated into the transformation equation, which will reverse the results such that a lower raw functioning score results in a higher percentage of the functioning score, corresponding with better and more healthy functioning. Subsequently, the median scores of all functioning scales were calculated.

Regarding the assessment of symptoms, the equation recommended in the manual is used to calculate the fatigue, nausea and vomiting, pain, dyspnea, insomnia, appetite, constipation, diarrhea, and financial difficulties scores separately. The lower raw symptoms score results in a lower transformed score which corresponds to lower levels of symptoms/problems. In order to unify the score interpretation and match that of the functioning score (with a lower raw score resulting in a higher transformed score, meaning better outcomes), the transformed score of symptoms was subtracted from 100. This will result in an interpretation such that a lower raw symptoms score results in a higher transformed symptoms score, which indicates better health and a lower level of problematic symptoms. Next, the median scores of all symptom scales were calculated.

With respect to global health assessment, given the raw score of each question and the transformation equation, higher raw scores of global health correspond to higher transformed scores, indicating better global health.

According to these calculations and the scores transformation, the median of functioning scale, the symptoms scale, and the global health scale, as well as the median of all three scales out of 100 (overall health scale), were calculated and compared between treatment groups.

### Statistical Analysis

Descriptive statistics were used for data analysis. Differences in HRQoL were estimated and compared using the Kruskal–Wallis test (median (IQR)), while frequencies and percentages were used for categorical variables. Any associations between clinical outcomes and patient criteria were determined by Chi-squared tests. Data analysis was performed using SPSS software, version 28 (IBM Corp., Armonk, NY, USA), and a *p*-value of <0.05 indicates statistically significant results.

## 3. Results

### 3.1. Demographic Characterization of the Patients

A total of 199 patients (184 patients on a single ICI and 15 on more than one ICI) were identified to be eligible for the study. A total of 82 of those receiving single therapy and 11 of those receiving more than one medication answered the call and completed the questionnaire, resulting in a total response rate of 46.7%.

Of the participants, 39.8%, 35.5%, and 12.9% were on pembrolizumab, nivolumab, and atezolizumab as single agents, respectively, whereas 7.5% and 4.3% were treated with multiple anti-PD-1 (pembrolizumab and nivolumab) and a combination of anti-PD-1+ anti-PD-L1 (two patients on pembrolizumab + durvalumab; one patient on pembrolizumab + atezolizumab; one patient on nivolumab + atezolizumab). The median age and body mass index (BMI) of participants were 61 years and 26 kg/m^2^, respectively, with no significant difference between treatment groups, although pembrolizumab-treated patients were the youngest and exhibited the highest BMI score compared to those in the other groups. Approximately 69% of the patients were males, and 36.5% presented with one comorbidity (either psychiatric, neurological, cardiovascular, endocrine, respiratory, hepatic, renal, or genitourinary) and had been exposed previously (67.7%) to either targeted therapy, chemotherapy, or a combination of both. A significant difference was found in the type of primary diagnosed cancer and stage at ICI initiation; specifically, solid malignancies and a higher stage of cancer were the most common characteristics in all groups. Further details about the baseline characteristics are provided in Table 1.

### 3.2. HRQoL of the Patients

Upon investigating patient functioning, the median score for all treatment groups was 84.7/100 (Table 2). Although a score > 80 was found when assessing role, cognitive, emotional, and social functioning in all combined groups (Table 2), as well as in most of the groups when they were compared together (Table 2 and Figure 1B–E), physical functioning tends to be the most affected, with a median score of 73.3/100 in all combined groups (Table 2), as well as in the separated groups, particularly for pembrolizumab and nivolumab (Table 2 and Figure 1A). This was clearly reflected by the reduced ability to participate in strenuous activities and to walk for long distances, as >70% have confirmed this effect (Table 3). Interestingly, patients who received a single therapy exhibited worse physical functioning than those receiving several anti-PD-1 and a combination of anti-PD-1 + anti-PD-L1 (Table 2 and Figure 1A) drugs. Despite these variations, no significant difference in either physical or other functioning scales was observed between treatment groups (Table 2 and Figure 1A–D). Notably, emotional functioning was the lowest among those treated with multiple anti-PD-1s, with a score of 75/100, compared with that of other treatment groups, yet it was not significant (Table 2 and Figure 1E). Cumulatively, the overall functioning score did not differ significantly between treatment groups (Table 2 and Figure 2A). Further information regarding the proportion of patients complaining of functional decline using all scales and the scores of the treatment groups are shown in Table 2 and Table 3 and Figure 1 and Figure 2.

In the assessment of symptoms, more than half of the patients complained of fatigue and pain (Table 3), which impacted the overall cohort’s score related to these symptoms, which was 88.8/100 and 83.3/100, respectively (Table 2). Interestingly, although there was no significant variation in the scores of all combined symptoms among all groups (*p* = 0.931) (Table 2 and Figure 2B), patients receiving anti-PD-1 + anti-PD-L1 seemed to have more complains about fatigue, pain, dyspnea, and constipation; hence they exhibited the worst, yet not significantly so, scores compared to those of the other treatment groups (Table 2 and Table 3). Further information regarding patients complaining of symptoms and the scores of the treatment groups is provided in Table 2 and Table 3.

Regarding the global health assessment, all combined groups scored > 80/100 (Table 2). Nevertheless, nivolumab-treated patients exhibited the worst, yet not significantly so, health compared to that of the other groups (Table 2 and Figure 2C). Further information regarding the scores for global health-related questions and the proportion of patients answering these questions is shown in Table 2 and Table 3.

Taken together, the estimated overall score based on all scales was 82.7/100, with no significant difference between treatment groups (Table 2 and Figure 2D).

## 4. Discussion

Although ICIs have been recently implemented in clinical practice, they have become the standard of care in patients diagnosed with several types of cancer, given their benefits in regards to disease remission and progression [29]. This widespread use of ICIs urges us to further investigate their impact on patient quality of life, whether used as primary, adjuvant, or second-line therapies. In this study, we prospectively investigated the HRQoL in real-world adult patients treated with different ICIs regimens in two tertiary, educational hospitals.

In our study, 39.8%, 35.5%, and 12.9% of patients received pembrolizumab, nivolumab, and atezolizumab as single agents, respectively, while smaller percentages were treated with multiple anti-PD-1 or combinations of anti-PD-1 and anti-PD-L1 therapies. This distribution reflects the current trends in immunotherapy treatments for cancer, particularly the widespread use of pembrolizumab and nivolumab compared to that of ipilimumab (anti-CTLA-4) due to their side-effect profiles and their proven efficacy for treatment of various types of cancer [30,31].

When investigating patient functioning (role, cognitive, emotional, etc.), we found comparable, non-significantly different levels among treatment groups. However, patients treated with single anti-PD-L1 and, in particular, anti-PD-1 agents, reported lower physical, role, and cognitive functioning scores than those treated with multiple anti-PD-1 and more specifically, combination therapy. Interestingly, patients treated with combination therapy tended to have better overall functioning, reflected by higher questionnaire scores than those treated with multiple anti-PD-1, single anti-PD-1, or anti-PD-L1 agents. This is in contrast to the results published by Looman et al., in which patients treated with combination therapy (ipilimumab/nivolumab) reported lower HRQoL scores compared to those treated with anti-PD-1 therapy [22]. This controversy could be attributed to the fact that the use of ipilimumab in the combination therapy investigated by Looman et al. has shifted toward pembrolizumab and nivolumab over ipilimumab, given the superiority of newer agents in regards to health-related outcomes [32]. Cumulatively, the overall scores of all functioning were reasonable reflected by a median score of 84.7/100 in all study cohorts.

Fatigue and pain were the most common symptoms, complained of by more than 50% of the cohort, yet no significant difference was found between treatment groups (*p* = 0.325, 0.239, and 0.642). These findings align with the results reported by Looman et al., in which physical fatigue was more frequently observed in their patients, with no difference between the single anti-PD-1 agent and the combination therapy groups [22]. Interestingly, although our results showed fatigue to be more prominent in groups treated with a single anti-PD-1 agent (45.5% and 59.5% with pembrolizumab and 67.6% and 51.5% with nivolumab) than in those treated with anti-PD-L1 and combination therapy groups, the complaint of pain tended to be more frequent in the combination group (75%). This certainly contributed to the decline observed in physical and role functioning scores for the single anti-PD-1 agents group compared to those of the combination group, ultimately impacting the overall functioning scores in these groups. Generally speaking, the overall scores of symptoms were acceptable reflected by a median score of 85.19/100 in all study cohorts.

Regarding global health assessment, the present study reported a reasonable score 83.3/100 for all combined groups, although nivolumab-treated patients exhibited the worst (75/100) global health, but it was not significantly different compared to that of the other groups. This is in line with results of Looman et al. regarding the overall global health score and more importantly, with the use of nivolumab, which can be explained by the higher prevalence of immune-related adverse effects (irAEs) during nivolumab treatment, consequently impacting the long-term HRQoL of these patients [22]. On the contrary, the findings from previous studies showed that the majority of patients (54.8%) reported poor global health, and only 17.4% perceiving their health to be good. This discrepancy between our results and those reported previously might be attributed to differences in patient populations, cancer stage at treatment initiation, treatment regimens, and assessment methods [23].

Patients in this study reported an acceptable overall health score of 82.7/100 associated with the use ICIs therapies, with no significant differences between treatment groups. This overall positive outlook contrasts with those reflected in earlier research, which often highlighted a significant decline in HRQoL domains such as role, social, and emotional functioning among patients undergoing immunotherapy [13,22,23]. This controversy could be attributed to the absence of a unified, recognized instrument recommended by international guidelines to evaluate HRQoL among ICIs-treated patients, which creates some concerns regarding the analysis, reporting, or interpretation of HRQoL among cancer patients [14,33,34]. Also, patient adaptation to cancer and its management could potentially influence HRQoL reporting at any level [14,35]. In other words, patient perception regarding the scenario of worsening of cancer and its therapy or the general efficacy of newer agents such ICIs could positively or negatively impact the reporting of HRQoL. Nevertheless, the improvements observed in the present study might reflect the advances in the management of adverse effects and supportive care that ameliorated the impact of ICIs on HRQoL.

The present study’s findings are consistent with the evolving landscape of immunotherapy, in which newer agents like pembrolizumab and nivolumab are increasingly preferred due to their superior efficacy and manageable side effect profiles [30,31]. The noted differences in physical and emotional functioning between single and multiple or combination therapy groups underscore the importance of personalized treatment plans that consider both the efficacy and the impact on quality of life of different immunotherapy regimens.

Several limitations exist in our study. First, despite the tremendous effort to include all eligible participants, a small sample size, particularly for the multiple anti-PD-1 and combination therapy groups, along with a low response rate of 47%, as well as the utilization of only two healthcare centers, limits the generalizability of the study’s findings. Therefore, multi-center studies are necessary to overcome these limitations. Second, the study provides a snapshot of HRQoL at a single time point, leaving the long-term effects of ICIs on HRQoL unclear. Longitudinal data provides better assessment of the sustainability of the reported outcomes. However, this could be challenged by the potential decline in the response rate when such an approach is used. Third, the heterogeneity of the treatment regimens received by the patients, in addition to the ICIs, complicates the isolation of the effects of individual treatments on HRQoL. Considering other types of cancer therapies as a factor in the analysis and employing several questionnaires may potentially reduce the impact of treatment heterogeneity on the assessment of HRQoL. Additionally, setting more stringent inclusion/exclusion criteria could reduce this effect, yet it may also compromise the sample size. Fourth, although the EORTC-QLQ-C30 questionnaire is designed to be used for cancer patients, it is still considered as non-specific for those treated with ICIs due to its limited assessment of irAEs associated with ICIs, which certainly impacts patient responses to questions assessing HRQoL. A new specific questionnaire (2054 Immune Checkpoints Inhibitor) is currently being developed [36], but it is not currently accessible, since it has not been fully validated. Fifth, the HRQoL assessments relied on self-reported data from patients, which can be biased by their perceptions and psychological states, introducing some variability in the findings. Guiding patients throughout completing the questionnaire would potentially ameliorate this bias. In spite of these limitations, our study is the first of its kind to be published using local data, adding more scientific value to the limited research published in this area.

## 5. Conclusions

Our study revealed potentially similar impact of ICIs, specifically anti-PD-1 and anti-PD-L1, on the health-related quality of life of cancer patients. Despite this fact, our findings align with the evolving preference for the use of pembrolizumab and nivolumab in clinical practice, reflected by their efficacy and manageable side-effect profiles. This study also highlights the need for continued research, especially the use of longitudinal studies and more diverse patient cohorts, to fully understand the long-term impacts of immunotherapy, particularly ICIs, on the health-related quality of life of cancer patients.

## Figures and Tables

**Figure 1 jcm-13-04918-f001:**
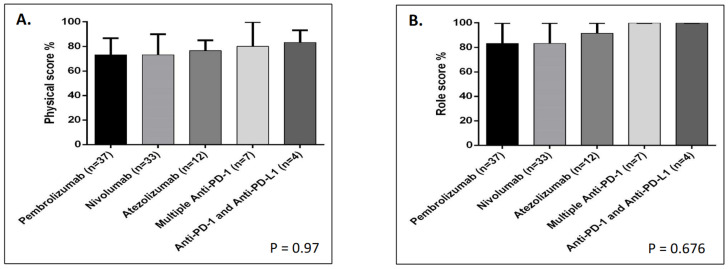
Comparison of functioning domains used for functional assessment in EORTC QLQ-C30 between treatment groups (n = 93). (**A**) Physical score %; (**B**) role score %; (**C**) cognitive score %; (**D**) social score %; (**E**) emotional score %.

**Figure 2 jcm-13-04918-f002:**
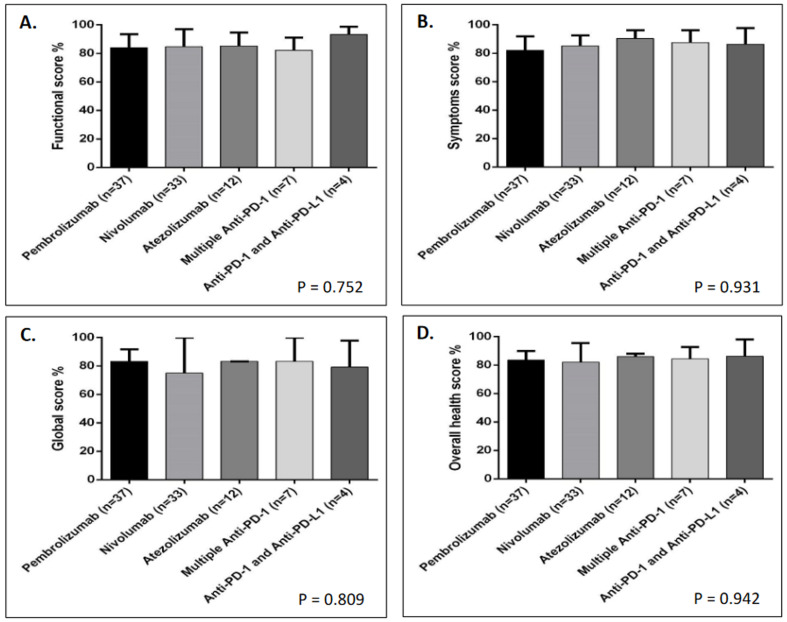
Comparison of scores between treatment groups (n = 93) based on (**A**) functional, (**B**) symptoms, (**C**) global health, and (**D**) overall health scales of EORTC QLQ-C30.

**Table 1 jcm-13-04918-t001:** Baseline characteristics of ICIs-treated patients.

Characteristics	All Patients (n = 93)	Pembrolizumab (n = 37)	Nivolumab (n = 33)	Atezolizumab (n = 12)	Multiple Anti-PD-1 (n = 7)	Anti-PD-1 + Anti-PD-L1 (n = 4)	*p*-Value
Age [median (IQR)]	61 (27)	47 (21)	62 (39)	66 (9.5)	62 (43)	59 (2.5)	0.373
BMI, kg/m^2^ [median (IQR)]	26 (8.6)	28 (12.1)	24.4 (9.1)	24.17 (3.9)	21.46 (7.2)	25.84 (2.1)	0.059
Gender, n (%) Female Male	29 (31.2) 64 (68.8)	17 (45.9) 20 (54.1)	7 (21.2) 26 (78.8)	2 (16.7) 10 (83.3)	3 (42.9) 4 (57.1)	0 4 (100)	0.068
Comorbidities, n (%) 0 1 2 3 4	30 (32.3) 34 (36.6) 21 (22.6) 6 (6.5) 2 (2.2)	8 (26.7) 10 (29.4) 15 (71.4) 3 (50) 1 (50)	17 (56.7) 10 (29.4) 3 (14.3) 2 (33.3) 1 (50)	2 (6.7) 7 (20.6) 2 (9.5) 1 (16.7) 0	3 (10) 3 (8.8) 1 (4.8) 0 0	0 4 (11.8) 0 0 0	0.059
Previous exposure to cancer therapy, n (%) Yes No	63 (67.7) 30 (32.3)	25 (67.6) 12 (32.4)	9 (27.3) 24 (72.7)	7 (58.3) 5 (41.7)	2 (28.6) 5 (71.4)	0 4 (100)	0.195
Primary diagnosed cancer, n (%) Solid Hematological	82 (88.2) 11 (11.8)	35 (94.6) 2 (5.4)	27 (81.8) 6 (18.2)	12 (100) 0	4 (57.1) 3 (42.9)	4 (100) 0	0.023
Cancer stage right before ICI initiation, n (%) Stage II Stage III Stage IV	12 (12.9) 17 (18.3) 64 (68.8)	3 (8.1) 4 (10.8) 30 (81.1)	3 (9.1) 10 (30.3) 20 (60.6)	5 (41.7) 0 7 (58.3)	0 2 (28.6) 5 (71.4)	1 (25) 1 (25) 2 (50)	0.021

**Table 2 jcm-13-04918-t002:** Scoring of each scale/item of EORTC QLQ-C30 questionnaire according to type of treatment, with reversed scoring of the symptom scale.

Variable [Median (IQR)]	All Treatment Groups (n = 93)	Pembrolizumab (n = 37)	Nivolumab (n = 33)	Atezolizumab (n = 12)	Multiple Anti-PD-1 (n = 7)	Anti-PD-1 + Anti-PD-L1 (n = 4)	*p*-Value
Physical functioning	73.3 (33.3)	73.3 (40)	73.3 (33.3)	76.6 (23.3)	80 (36.7)	83.3 (33.3)	0.97
Role functioning	83.3 (33.3)	83.3 (50)	83.3 (33.3)	91.7 (33.3)	100 (33.3)	100 (12.5)	0.676
Cognitive functioning	100 (33.3)	83.3 (33.3)	83.3 (33.3)	100 (4.1)	100 (25)	91.7 (20.8)	0.466
Emotional functioning	100 (20.8)	91.7 (16.7)	100 (16.7)	100 (33.3)	75 (20.8)	100 (2.1)	0.269
Social functioning	100 (16.6)	100 (16.7)	100 (0)	100 (33.3)	100 (25)	100 (0)	0.475
All functioning	84.7 (19.8)	84 (23.3)	84.7 (21)	85.3 (11.17)	82.3 (7.8)	93.3 (15.4)	0.752
Fatigue	88.8 (33.3)	77.8 (16.7)	88.9 (16.7)	100.0 (4.16)	88.9 (16.7)	77.8 (0)	0.234
Nausea and vomiting	100 (16.7)	100.0 (33.3)	100.0 (33.3)	100.0 (33.3)	100.0 (25)	100.0 (37.5)	0.536
Pain	83.3 (33.3)	83.3 (33.3)	83.3 (33.3)	83.3 (33.3)	83.3 (33.3)	66.7 (75)	0.791
Dyspnea	100 (33.3)	100.0 (33.3)	100.0 (33.3)	100.0 (41.6)	100.0	66.7 (16.7)	0.704
Insomnia	100 (33.3)	100.0 (33.3)	100.0 (33.3)	83.3 (33.3)	100.0 (16.7)	100.0 (25)	0.531
Appetite	100 (33.3)	100.0 (33.3)	66.7 (33.3)	66.7 (41.6)	100.0 (16.7)	100.0 (66.7)	0.874
Constipation	100 (33.3)	100.0 (0)	100.0 (33.3)	100.0 (0)	100.0 (0)	66.7 (0)	0.86
Diarrhea	100 (0)	100.0 (0)	100.0 (0)	100.0 (0)	100.0 (16.7)	100.0 (0)	0.605
Financial	100 (0)	100.0 (16.7)	100.0 (16.7)	100.0 (4.16)	100.0 (16.7)	100.0 (0)	0.434
All symptoms	85.19 (19.1)	82.1 (17.9)	85.18 (19.1)	90.4 (24.2)	87.7 (20.4)	86.4 (19)	0.931
Global health	83.3 (25)	83.3 (33.3)	75 (33.3)	83.3 (16.7)	83.3 (16.7)	79.17 (29.17)	0.809
Overall health	82.7 (17.6)	83.5 (20.4)	82.1 (21.2)	86.0 (12)	84.4 (14.3)	86.3 (21.3)	0.948

**Table 3 jcm-13-04918-t003:** Response to EORTC-QLQ-C30 items in each scale according to type of treatment.

Question	All Treatment Groups (n = 93)	Pembrolizumab (n = 37)	Nivolumab (n = 33)	Atezolizumab (n = 12)	Multiple Anti-PD-1 (n = 7)	Anti-PD-1 + Anti-PD-L1 (n = 4)	*p*-Value
Functional scale							
Do you have any trouble doing strenuous activities like carrying a heavy shopping bag or a suitcase? Yes (n,%)	67 (72)	28 (75.7)	23 (69.7)	8 (66.7)	5 (71.4)	3 (75)	0.97
Do you have any trouble taking a long walk? Yes (n,%)	68 (73.1)	30 (81.1)	21 (63.3)	9 (75)	5 (71.4)	3 (75)	0.602
Do you have any trouble taking a short walk outside of the house? Yes (n,%)	29 (31.2)	14 (37.8)	10 (30.3)	1 (8.3)	3 (42.9)	1 (25)	0.378
Do you need to stay in bed or a chair during the day? Yes (n,%)	62 (66.7)	26 (70.3)	22 (66.7)	8 (66.7)	4 (57.1)	2 (50)	0.91
Do you need help with eating, dressing, washing yourself or using the toilet? Yes (n,%)	14 (15.1)	6 (16.2)	5 (15.2)	1 (8.3)	1 (14.3)	1 (25)	0.942
During the past week, were you limited in doing either your work or other daily activities? Yes (n,%)	44 (47.3)	20 (54.1)	15 (45.5)	5 (41.7)	3 (42.9)	1 (25)	0.786
During the past week, were you limited in pursuing your hobbies or other leisure time activities? Yes (n,%)	46 (49.5)	20 (54.1)	16 (48.5)	6 (50)	3 (42.9)	1 (25)	0.843
During the past week, have you had difficulty in concentrating on things, like reading a newspaper or watching television? Yes (n,%)	27 (29)	13 (35.1)	9 (27.3)	1 (8.3)	2 (28.6)	2 (50)	0.397
During the past week, have you had difficulty remembering things? Yes (n,%)	38 (40.9)	18 (48.6)	14 (42.4)	2 (16.7)	3 (42.9)	1 (25)	0.367
During the past week, did you feel tense? Yes (n,%)	27 (29)	12 (32.4)	5 (15.2)	5 (41.7)	5 (71.4)	0	0.018
During the past week, did you worry? Yes (n,%)	30 (32.3)	14 (37.8)	7 (21.2)	5 (41.7)	4 (57.1)	0	0.150
During the past week, did you feel irritable? Yes (n,%)	29 (31.2)	9 (24.3)	13 (39.4)	3 (25)	4 (57.1)	0	0.194
During the past week, did you feel depressed? Yes (n,%)	25 (26.9)	6 (16.2)	11 (33.3)	3 (25)	4 (57.1)	1 (25)	0.19
During the past week, has your physical condition or medical treatment interfered with your family life? Yes (n,%)	16 (17.2)	6 (16.2)	4 (12.2)	4 (33.3)	2 (28.6)	0	0.369
During the past week, has your physical condition or medical treatment interfered with your social activities? Yes (n,%)	21 (22.6)	8 (21.6)	6 (18.2)	4 (33.3)	3 (42.9)	0	0.407
Symptoms scale							
During the past week, have you felt weak? Yes (n,%)	37 (39.8)	15 (40.5)	15 (45.5)	2 (16.7)	3 (42.9)	2 (50)	0.504
During the past week, were you tired? Yes (n,%)	45 (48.4)	22 (59.5)	15 (45.5)	3 (25)	3 (42.9)	2 (50)	0.325
During the past week, did you need to rest? Yes (n,%)	53 (57)	25 (67.6)	17 (51.5)	4 (33.3)	5 (71.4)	2 (50)	0.239
During the past week, have you felt nauseated? Yes (n,%)	33 (35.5)	14 (37.8)	13 (39.4)	3 (25)	3 (42.9)	0	0.517
During the past week, have you vomited? Yes (n,%)	18 (19.4)	9 (24.3)	6 (18.2)	2 (16.7)	1 (14.3)	0	0.782
During the past week, have you had pain? Yes (n,%)	51 (54.8)	23 (62.2)	16 (48.5)	6 (50)	3 (42.9)	3 (75)	0.642
During the past week, did pain interfere with your daily activities? Yes (n,%)	39 (41.9)	15 (40.5)	16 (48.5)	4 (33.3)	3 (42.9)	1 (25)	0.836
During the past week, were you short of breath? Yes (n,%)	35 (37.6)	12 (32.4)	14 (42.4)	4 (33.3)	3 (42.9)	2 (50)	0.88
During the past week, have you had trouble sleeping? Yes (n,%)	40 (43)	18 (48.6)	14 (42.4)	6 (50)	1 (14.3)	1 (25)	0.461
During the past week, have you lacked appetite? Yes (n,%)	45 (48.4)	18 (48.6)	17 (51.5)	7 (58.3)	2 (28.6)	1 (25)	0.630
During the past week, have you been constipated? Yes (n,%)	39 (41.9)	14 (37.8)	16 (48.5)	5 (41.7)	2 (28.6)	2 (50)	0.834
During the past week, have you had diarrhea? Yes (n,%)	17 (18.3)	5 (13.5)	9 (27.3)	2 (16.7)	1 (14.3)	0	0.503
During the past week, has your physical condition or medical treatment caused you financial difficulties? Yes (n,%)	11 (11.8)	5 (13.5)	2 (6.1)	2 (16.7)	2 (28.6)	0	0.428
Global scale							
During the past week, how would you rate your overall health during the past week? [median score (IQR)]	6 (2)	5 (2)	5 (2)	6 (1.25)	6 (1)	5.5 (1.5)	0.779
During the past week, how would you rate your overall quality of life during the past week? [median score (IQR)]	6 (2)	6 (2)	6 (2)	6 (1)	6 (1)	6 (2)	0.688

## Data Availability

Data presented in this study are available upon request from the corresponding author. Data is not publicly available due to privacy reasons.

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
