# Peer review of "Real-World Comparison of Health-Related Quality of Life Associated with Use of Immune-Checkpoint Inhibitors in Oncology Patients"

_jcm, 2024, doi:10.3390/jcm13164918_

Round 1

Reviewer 1 Report

Comments and Suggestions for Authors

Spelling correction in table 1, Pembrolizuma

No significant differences in the quality of life (QOL) was observed among ICI treated patients. Anti-CTLA4 remains an exception. This is a report from prospective questionnaire-based study is Saudi Arabia. 93 out of 199 patients completed the study. Nivolumab- treated patients showed the worst global health, although it was not statistically significant.  Several variable functions were tested to prepare the detail report.

Comments on the Quality of English Language

Spelling correction in table 1, Pembrolizuma

No significant differences in the quality of life (QOL) was observed among ICI treated patients. Anti-CTLA4 remains an exception. This is a report from prospective questionnaire-based study is Saudi Arabia. 93 out of 199 patients completed the study. Nivolumab- treated patients showed the worst global health, although it was not statistically significant.  Several variable functions were tested to prepare the detail report.

Author Response

We appreciate the reviewer for their time to review and comment on our manuscript. Below is our response to each comment. All modifications to the manuscript are tracked.

Comment #1: Minor editing of English language required

Response: Minor English language edits, including grammatical and wording corrections, were done to the manuscript.

Comment #2: Spelling correction in table 1, Pembrolizuma

Response: Spelling was corrected as requested.

Reviewer 2 Report

Comments and Suggestions for Authors

The manuscript, entitled "Real-world Comparison of Health-related Quality of Life associated with use of Immune-Checkpoint Inhibitors in oncology patients" has been reviewed. 

This paper presents a prospective study based on quality of life questionnaires to evaluate the impact of the use of different immune checkpoint inhibitors (ICIs) on the health-related quality of life (HRQoL) of cancer patients.  In order to assess quality of life, the EORTC-QLQ-C30 questionnaire was selected for use in this study, as it is one of the most reliable and valid instruments currently available.The study population consisted of oncology patients from two distinct health centers, KKUH and NGH, who had received at least one dose of PD1/PD-L1 inhibitors. The selection of patients was appropriate and the information collected in the questionnaires was correctly processed and analyzed. Furthermore, additional reports and literature studies have been provided to substantiate these results. The findings of this study highlight the necessity for further research into the HRQoL associated with the use of ICIs. This is particularly important in light of the increasing use of ICIs in a range of different pathologies and the current lack of consensus in the literature on this topic. Nevertheless, in view of the limitations of the present study, which were correctly identified, further studies are required to gain a deeper understanding and verify the findings.

This manuscript is in accordance with the scope of the journal. The English language used is appropriate for this publication. Further revisions may be necessary for the manuscript to comply with the final publication requirements. This includes recommendations on formatting and wording, as well as the finalization of specific ideas. 

1) Revision of wording
A. Abbreviations error It is advisable to provide the full name of the abbreviation or acronym in an early section of the main text. This helps to avoid any potential misunderstandings or misinterpretations. 1. The abbreviations for the "European Organization for Research and Treatment of Cancer" and the "Quality of Life Questionnaire" were not entered correctly in the main body of the manuscript. It is recommended that this information be included in the "Introduction" section, given the number of times it appears. 2. On line 63, the acronym PROMs is incorrectly defined. The word "outcomes" should be modified to its singular form. 3. The term "FACT" is employed on several occasions (lines 78, 122, 137). However, it is not correctly identified as "Functional Assessment of Cancer Therapy". It is thus recommended that this be revised to enhance comprehension, particularly for the non-expert reader. 4. The term "EQ-5D" is employed on two occasions (lines 78 and 106). "EuroQol 5 Dimensions" was not correctly referenced in the text. It is recommended that this be corrected to enhance comprehension of the paper, particularly for the general reader. 5. The term "FKSI-DRS" is used on two occasions (lines 87 and 88). In the text, it was not properly designated as “Functional Assessment of Cancer Therapy Kidney Cancer Symptom Index – Disease Related Symptoms”. It is recommended that this be revised to enhance comprehension of the text, particularly for the reader without a background in this field. 6. Line 89, the acronym “HRQOL” should be corrected to “HRQoL”. B. Grammatical errors 1. Line 78, the acronym FACT-Ga should be removed from the parentheses. 2. Line 97, it is more accurate to utilize the term "improvement" instead of "improved". 3. Line 146, the phrase "questionnaire based-study" should be corrected to "questionnaire-based study". 4. Line 158, the phrase "Multi-items scales ..." should be corrected to "Multi-item scales ...". 5. Line 185, the phrase “... was subtracted form 100” should be corrected to “... was subtracted from 100”. 6. Lines 187-188, the subject "median scores" is plural; thus, the verb should be in the plural form, "were calculated," rather than the singular form, "was calculated". 7. In line 196, the subject "differences" is plural; thus, the verb should be "were estimated" rather than "was estimated". 8. Line 210, the sentence was concluded with a semicolon, which is an incorrect punctuation mark. It should be replaced with a period. 9. Line 322, the appropriate term is "acceptable" instead of "accepted". 10. Line 348, the sentence was concluded with a semicolon, which is an incorrect punctuation mark. It should be replaced with a period. C. Clarity error 1. Line 166, the word "except" should be replaced with "exception". 2. It is recommended that paragraphs 173-178 be modified to provide greater clarity and understanding, thereby avoiding misinterpretation by the reader. 3. Line 205-206, an error has been identified in the sum of the "single therapy". The correct value is 82. 4. In order to enhance clarity and avoid misinterpretation, it would be beneficial to modify paragraphs 329-331. It would be advisable to rewrite the phrase "patient's adaptation of their cancer disease."

2) Revision of the format A. Text 1. It is advisable to adopt a unified format for referencing studies by other authors. In the "Introduction" section, the format "... and colleagues" is utilized, whereas the "Discussion" section employs the format "... et al." 2. It is recommended that citations from websites and online resources (lines 156, 173, 358) be referenced in the "References" section instead of appearing in the main text, in order to maintain the correct format of the manuscript. The optimal format for citation is as follows: Author (if available). Title of the webpage (if available). Available online: http://... (accessed on date). 3. The use of underlined text for emphasis within the manuscript is discouraged. Underlined text has been found on line 112. 4. The use of bold text for emphasis within the manuscript is discouraged. Bold text has been identified in the following lines: 173-174, 179, 189, 189, 227, 254, 263, 268, 285, 299, 312. Furthermore, it has been identified in the “References” section, specifically in the volume number of the articles listed. 5. It is recommended that capital letters be used to refer to figures and tables in the main text. The following lines require revision: 222, 228, 229, 230, 231, 232, 233, 234, 236, 238, 240, 241, 242, 244, 255, 256, 257, 260, 261, 262, 263, 265, 267, 269. 6. The bibliographic reference number 5 must be revised in accordance with the formatting requirements of the journal. B. Tables and Figures 1. It is recommended that the columns in each table be adjusted in order to ensure accurate reading of the names. 2. Table 1 contains the following errors, which require correction: a. The “Comorbidities: 3” row for the “Atezolizumab” column is missing a parenthesis in the corresponding IQR values. b. In the "Comorbidities" row, the values placed in parentheses have not been correctly identified as percentages. c. An asterisk has been included in the row "Cancer stage right..." but the purpose of this addition has not been explicitly defined. d. It is advisable to correctly separate Table 1 from the main text. 3. In "Table 3", for the "Question" column, it is recommended to use justified alignment. This is usually more correct for long sentences. 4. In the description of the figures, it is advisable to include the identification of each of the graphs grouped.
3) Revision of concepts and finalization of ideas A. In the “Introduction” section, it is recommended to add a short line describing the importance and impact of ICIs in cancer treatment. It is also important to briefly describe the ICIs used in this study (PD1 and PD-L1). This information would be very useful for the non-expert reader. B. The information provided in the first paragraphs of the "Results" section, lines 204 - 213, corresponds more closely to the information that should be included in the "Methods" section. This is because it describes the patient recruitment process, the number of patients, and the classification into specific groups. C. When the sample size is small, it is important to use careful language when discussing the results. This is the case for the "multiple anti-PD1 therapy" and "combination therapy" groups.It is recommended to use expressions in potential, to emphasize trends rather than direct statements, and to avoid generalizations. The "Discussion" section should also include a short paragraph commenting on these limitations. D. In the "Conclusions" section, the following is mentioned: "... our results showed no significant differences between the different treatment regimens, with the exception of CTLA-4 inhibitors, ...". In the context of the study conducted by the authors, the experimental design does not include the evaluation of anti-CTLA-4 ICIs, making this clarification meaningless. It is recommended to rewrite this paragraph. E. It is recommended to add a paragraph in the "Conclusions" section to reiterate and consolidate the final message that has been built up during the discussion and to emphasize the objective pursued by the study. It is important to highlight the clinical implications as well as the importance of conducting this type of study to achieve a better understanding of the quality of life of cancer patients treated with this type of treatment.

Author Response

We appreciate the reviewer for their time to review and comment on our manuscript. Below is our response to each comment. All modifications to the manuscript are tracked.

Comment #1: English language fine. No issues detected

Response: Minor English language edits, including grammatical and wording corrections, were done to the manuscript as one of the reviewers requested that.

Comment #2: This paper presents a prospective study based on quality of life questionnaires to evaluate the impact of the use of different immune checkpoint inhibitors (ICIs) on the health-related quality of life (HRQoL) of cancer patients.  In order to assess quality of life, the EORTC-QLQ-C30 questionnaire was selected for use in this study, as it is one of the most reliable and valid instruments currently available. The study population consisted of oncology patients from two distinct health centers, KKUH and NGH, who had received at least one dose of PD1/PD-L1 inhibitors. The selection of patients was appropriate and the information collected in the questionnaires was correctly processed and analyzed. Furthermore, additional reports and literature studies have been provided to substantiate these results. The findings of this study highlight the necessity for further research into the HRQoL associated with the use of ICIs. This is particularly important in light of the increasing use of ICIs in a range of different pathologies and the current lack of consensus in the literature on this topic. Nevertheless, in view of the limitations of the present study, which were correctly identified, further studies are required to gain a deeper understanding and verify the findings.

Response: We totally agree that the study highlighted the necessity for further research into the HRQoL associated with the use of ICIs given the increase use of ICI in a range of different pathologies and the current lack of consensus in the literature on this topic

Comment #3: This manuscript is in accordance with the scope of the journal. The English language used is appropriate for this publication. Further revisions may be necessary for the manuscript to comply with the final publication requirements. This includes recommendations on formatting and wording, as well as the finalization of specific ideas.

Response: As mentioned earlier, minor English language edits including grammatical and wording corrections were done to the manuscript as one of the reviewers requested that.

Comment #4: 1) Revision of wording

a. Abbreviations error

It is advisable to provide the full name of the abbreviation or acronym in an early section of the main text. This helps to avoid any potential misunderstandings or misinterpretations.

Comment #4a1: The abbreviations for the "European Organization for Research and Treatment of Cancer" and the "Quality of Life Questionnaire" were not entered correctly in the main body of the manuscript. It is recommended that this information be included in the "Introduction" section, given the number of times it appears.

Response: All abbreviations related to questionnaires and ICI classes in the introduction have been included and corrected as requested.

Comment #4a2: On line 63, the acronym PROMs is incorrectly defined. The word "outcomes" should be modified to its singular form.

Response: The word "outcomes" was modified to its singular form.

Comment #4a3: The term "FACT" is employed on several occasions (lines 78, 122, 137). However, it is not correctly identified as "Functional Assessment of Cancer Therapy". It is thus recommended that this be revised to enhance comprehension, particularly for the non-expert reader.

Response: It is “the Functional Assessment of Cancer Therapy”. This was included in the text.

Comment #4a4: The term "EQ-5D" is employed on two occasions (lines 78 and 106). "EuroQol 5 Dimensions" was not correctly referenced in the text. It is recommended that this be corrected to enhance comprehension of the paper, particularly for the general reader.

Response: It is “the EuroQol instrument-5 dimensions”. This was included in the text.

Comment #4a5: The term "FKSI-DRS" is used on two occasions (lines 87 and 88). In the text, it was not properly designated as “Functional Assessment of Cancer Therapy Kidney Cancer Symptom Index – Disease Related Symptoms”. It is recommended that this be revised to enhance comprehension of the text, particularly for the reader without a background in this field.

Response: It is “the Functional Assessment of Cancer Therapy–Kidney Symptom Index Disease-Related Symptoms”. This was included in the text.

Comment #4a6: Line 89, the acronym “HRQOL” should be corrected to “HRQoL”.

Response: The abbreviation “HRQOL” was corrected as requested.

b. Grammatical errors

Comment #4b1: Line 78, the acronym FACT-Ga should be removed from the parentheses.

Response: Since full name of the questionnaire “the Functional Assessment of Cancer Therapy-Gastric” was included in the text as suggested above, the abbreviation is preferred to appear in parentheses (FACT-Ga).

Comment #4b2: Line 97, it is more accurate to utilize the term "improvement" instead of "improved".

Response: It was changed as recommended.

Comment #4b3: Line 146, the phrase "questionnaire based-study" should be corrected to "questionnaire-based study".

Response: It was corrected as recommended.

Comment #4b4: Line 158, the phrase "Multi-items scales ..." should be corrected to "Multi-item scales ...".

Response: It was corrected as recommended.

Comment #4b5: Line 185, the phrase “... was subtracted form 100” should be corrected to “... was subtracted from 100”.

Response: It was corrected as recommended.

Comment #4b6: Lines 187-188, the subject "median scores" is plural; thus, the verb should be in the plural form, "were calculated," rather than the singular form, "was calculated".

Response: It was corrected as recommended.

Comment #4b7: In line 196, the subject "differences" is plural; thus, the verb should be "were estimated" rather than "was estimated".

Response: It was corrected as recommended.

Comment #4b8: Line 210, the sentence was concluded with a semicolon, which is an incorrect punctuation mark. It should be replaced with a period.

Response: It was corrected as recommended.

Comment #4b9: Line 322, the appropriate term is "acceptable" instead of "accepted".

Response: It was changes as recommended.

Comment #4b10: Line 348, the sentence was concluded with a semicolon, which is an incorrect punctuation mark. It should be replaced with a period.

Response: It was changes as recommended.

c. Clarity error

Comment #4c1: Line 166, the word "except" should be replaced with "exception".

Response: It was changes as recommended.

Comment #4c2: It is recommended that paragraphs 173-178 be modified to provide greater clarity and understanding, thereby avoiding misinterpretation by the reader.

Response: It was modified as recommended to avoid confusing the reader.

Comment #4c3: Line 205-206, an error has been identified in the sum of the "single therapy". The correct value is 82.

Response: It was corrected as requested.

Comment #4c4: In order to enhance clarity and avoid misinterpretation, it would be beneficial to modify paragraphs 329-331. It would be advisable to rewrite the phrase "patient's adaptation of their cancer disease."

Response: It was modified as requested to add mote clarity.

Comment #5: 2) Revision of the format

a. Text

Comment #5a1: It is advisable to adopt a unified format for referencing studies by other authors. In the "Introduction" section, the format "... and colleagues" is utilized, whereas the "Discussion" section employs the format "... et al."

Response: The “Author and colleagues…” format was adopted and used through all section of the manuscript.

Comment #5a2: It is recommended that citations from websites and online resources (lines 156, 173, 358) be referenced in the "References" section instead of appearing in the main text, in order to maintain the correct format of the manuscript. The optimal format for citation is as follows: Author (if available). Title of the webpage (if available). Available online: http://... (accessed on date).

Response: Website were cited using the suggested format and references rearrangement was performed accordingly.

Comment #5a3: The use of underlined text for emphasis within the manuscript is discouraged. Underlined text has been found on line 112.

Response: Underlining the text was removed as requested.

Comment #5a4: The use of bold text for emphasis within the manuscript is discouraged. Bold text has been identified in the following lines: 173-174, 179, 189, 189, 227, 254, 263, 268, 285, 299, 312. Furthermore, it has been identified in the “References” section, specifically in the volume number of the articles listed.

Response: Bolding the volume number of references was removed as requested.

Comment #5a5: It is recommended that capital letters be used to refer to figures and tables in the main text. The following lines require revision: 222, 228, 229, 230, 231, 232, 233, 234, 236, 238, 240, 241, 242, 244, 255, 256, 257, 260, 261, 262, 263, 265, 267, 269. 6. The bibliographic reference number 5 must be revised in accordance with the formatting requirements of the journal.

Response: The word of “FIGURE and TABLE” were capitalized as requested. Reference 5 was corrected as recommended.

b. Tables and Figures

Comment #5b1: It is recommended that the columns in each table be adjusted in order to ensure accurate reading of the names.

Response: Columns were adjusted to ensure accurate reading of the drug names.

Comment #5b2: Table 1 contains the following errors, which require correction:

Comment #5b2a: The “Comorbidities: 3” row for the “Atezolizumab” column is missing a parenthesis in the corresponding IQR values.

Response: Missing parenthesis was added to the corresponding % values.

Comment #5b2b: In the "Comorbidities" row, the values placed in parentheses have not been correctly identified as percentages.

Response: The values placed in parentheses have been identified as percentages.

Comment #5b2c: An asterisk has been included in the row "Cancer stage right..." but the purpose of this addition has not been explicitly defined.

Response: The asterisk indicates p < 0.05 which should have appeared in the row “Primary diagnosed...and Cancer stage right...". Since p values were included in Table 1, asterisk was removed.

Comment #5b2d: It is advisable to correctly separate Table 1 from the main text.

Response: Table 1 is already separated from the text.

Comment #5b3: In "Table 3", for the "Question" column, it is recommended to use justified alignment. This is usually more correct for long sentences.

Response: Justified alignment was used.

Comment #5b4: In the description of the figures, it is advisable to include the identification of each of the graphs grouped.

Response: Identification of groups were added to the figure description.

Comment #6: 3) Revision of concepts and finalization of ideas

Comment #6a: it is recommended to add a short line describing the importance and impact of ICIs in cancer treatment. It is also important to briefly describe the ICIs used in this study (PD1 and PD-L1). This information would be very useful for the non-expert reader.

Response: Regarding the importance and impact of ICIs on cancer treatment, it has been highlighted in the 1st paragraph of introduction (Please refer to lines 45-50 and 55-61). With respect of the information about PD-1 and PD-L1 information, a few statements were added in the same paragraph (lines 50-55).

Comment #6b: The information provided in the first paragraphs of the "Results" section, lines 204 - 213, corresponds more closely to the information that should be included in the "Methods" section. This is because it describes the patient recruitment process, the number of patients, and the classification into specific groups.

Response: We think it fits more to the results section as such numbers and details usually appear in the result section of the abstract. Despite that, a few statements were removed from the 1st section of the results to avoid such information duplication.

Comment #6c: When the sample size is small, it is important to use careful language when discussing the results. This is the case for the "multiple anti-PD1 therapy" and "combination therapy" groups. It is recommended to use expressions in potential, to emphasize trends rather than direct statements, and to avoid generalizations. The "Discussion" section should also include a short paragraph commenting on these limitations.

Response: All suggestions on the discussion and study limitations were incorporated in the text.

Comment #6d: In the "Conclusions" section, the following is mentioned: "... our results showed no significant differences between the different treatment regimens, with the exception of CTLA-4 inhibitors, ...". In the context of the study conducted by the authors, the experimental design does not include the evaluation of anti-CTLA-4 ICIs, making this clarification meaningless. It is recommended to rewrite this paragraph.

Response: Conclusion has been modified as requested.

Comment #6e: It is recommended to add a paragraph in the "Conclusions" section to reiterate and consolidate the final message that has been built up during the discussion and to emphasize the objective pursued by the study. It is important to highlight the clinical implications as well as the importance of conducting this type of study to achieve a better understanding of the quality of life of cancer patients treated with this type of treatment.

Response: Conclusion has been modified as requested. 

Reviewer 3 Report

Comments and Suggestions for Authors

The paper is well-constructed and comprehensive in its approach and findings. The introduction, research design, methods, results, and conclusions are appropriately handled and presented. Minor improvements are suggested to be more robust and impactful, providing clearer insights and stronger contributions to the field.

1. Although your introduction is thorough, it would be beneficial to include more up-to-date studies to ensure that the latest research is represented.

2. Identify the specific gaps in the literature that your study addresses to better position its novelty.

3. Provide a more explicit justification for the selected sample size, including any computations or considerations that influenced the determination of the number of participants.

4. Provide a comprehensive explanation of the statistical approaches employed in your analysis. This encompasses the justification for choosing particular tests and their suitability for your data.

5. Analyze and acknowledge potential biases in the selection of patients and describe the measures taken to minimize their impact. It is advisable to incorporate a sensitivity analysis to showcase the resilience of your findings.

6. Discuss the clinical implications of your findings. How can your results influence clinical practice or future research directions?

7. Discuss your study's limitations. Discuss how these limitations might affect the interpretation of your results and suggest ways future research can overcome these challenges.

Author Response

We appreciate the reviewer for their time to review and comment on our manuscript. Below is our response to each comment. All modifications to the manuscript are tracked.

Comment #1: English language fine. No issues detected

Response: Minor English language edits, including grammatical and wording corrections, were done to the manuscript as one of the reviewers requested that.

Comment #2: The paper is well-constructed and comprehensive in its approach and findings. The introduction, research design, methods, results, and conclusions are appropriately handled and presented. Minor improvements are suggested to be more robust and impactful, providing clearer insights and stronger contributions to the field.

Response: We appreciate the reviewer for this comment.

Comment #3: Although your introduction is thorough, it would be beneficial to include more up-to-date studies to ensure that the latest research is represented.

Response: Please note that there are few studies investigating this topic and all of them, based on our research, were included in the introduction. Looking at the references used in the introduction and method, 16 were articles published between 2020 and 2024, 6 between 2000 and 2020, 3 between 1900 and 2000, 2 websites, and 1 the questionnaire reference.

Comment #4: Identify the specific gaps in the literature that your study addresses to better position its novelty.

Response: We have mentioned the following statement in the last paragraph of introduction “Beside the scarcity of research globally and conflicting evidence of HRQoL associated with the use of ICIs in cancer treatment…”. This would highlight the novelty of the study worldwide.

Comment #5: Provide a more explicit justification for the selected sample size, including any computations or considerations that influenced the determination of the number of participants.

Response: Given the low number of patients on these medications, we have basically included all of adults who were a life and consented to participate in the study.

Comment #6: Provide a comprehensive explanation of the statistical approaches employed in your analysis. This encompasses the justification for choosing particular tests and their suitability for your data.

Response: Since patients were not normally distributed, non-parametric Kruskal–Wallis test was used for continuous variables analysis including baseline variables and questionnaire scores. For categorial analysis and studying any difference between treatment groups and its association with patients’ criteria, chi-squared test was employed.

Comment #7: Analyze and acknowledge potential biases in the selection of patients and describe the measures taken to minimize their impact. It is advisable to incorporate a sensitivity analysis to showcase the resilience of your findings.

Response: As clarified previously, we included all of adults who were a life and consented to participate in the study. The only issue was disturbance in distribution of patients among treatment groups, thus non-parametric test was used to overcome that.

Comment #8: Discuss the clinical implications of your findings. How can your results influence clinical practice or future research directions?

Response: In terms of the clinical practice, more emphasis on counseling patients regarding the negative symptoms, particularly fatigue and pain, is necessary and should be provided equally to those receiving anti-PD-1 or anti-PD-L1, single or combination therapy. Yet it should be cleared to the patients that they may potentially experience that to lower extent compared to what has been seen with chemotherapy, if previously used. With respect to future research direction, more compelling evidence is necessary to compare all ICIs, including anti-CTLA-4 therapies. For our future research direction, we currently work on another ICI-related project that certainly would increase the awareness of healthcare providers about these therapies.

Comment #9: Discuss your study's limitations. Discuss how these limitations might affect the interpretation of your results and suggest ways future research can overcome these challenges.

Response: More limitations were added and discussed as recommended.

Reviewer 4 Report

Comments and Suggestions for Authors

Alwhaibi A, et al. performed a prospective cohort study on ICI treated patients with advanced malignancies. There are several concerns: 

1. Since there is no pre-study sample size calculation, it is unclear whether those comparisons mentioned in the study have enough power. 

2. The authors should not report P values alone, more important things like estimates, 95% CIs should be included, P-values can be omitted in deed. 

3. In order to make a statement, the authors should include non-ICI treated patients. But this is impossible per study design.

4. Given the limited sample size, I do not think any conclusions can be reached. 

Author Response

We appreciate the reviewer for their time to review and comment on our manuscript. Below is our response to each comment. All modifications to the manuscript are tracked.

Comment #1: Since there is no pre-study sample size calculation, it is unclear whether those comparisons mentioned in the study have enough power.

Response: There has been no study investigated this area. Thus, we elected to include all patients receiving ICIs at the time of conducting this study at KKUH and NGH.

Comment #2: The authors should not report P values alone, more important things like estimates, 95% CIs should be included, P-values can be omitted in deed.

Response:. Since no significant changes were found in the results, we did not include estimates in the abstracts given word number count decided by the journal. Despite that, we have included estimates in the abstract as requested.

Comment #3: In order to make a statement, the authors should include non-ICI treated patients. But this is impossible per study design.

Response: As you mentioned, the design of study is restricted to those receiving ICIs only.

Comment #4: Given the limited sample size, I do not think any conclusions can be reached.

Response: We have highlighted several limitations related to our study including the generalizability, which indeed caused by the sample size issue.